# Utilization of the posterior iliac line for visualizing posterior column screws in obturator oblique view

Hongtao Li[1], Li Xu[1], Longxin An[2], Xiaojing Li[1], Linjing Zhang[1], Jun Liu[1☯*], Kaili Zhai[3☯*], Xuecheng Sun[1☯*], Naibo Feng📵[1☯*]

1 Department of Trauma Orthopedics, Weifang People's Hospital, Shandong Second Medical University, Weifang, China, 2 Shandong Second Medical University, Weifang, China, 3 School of Cyberspace Security, Shandong Vocational College of Science and Technology, Weifang, China

☯ These authors contributed equally to this work.
* ljwfrmyy@sina.com (JL); 842460393@qq.com (KZ); 15853637565@163.com (XS); dr.feng1010@gmail.com (NF)

## Abstract

### Purpose

To evaluate whether posterior column screws penetrate the posterior cortical surface of the acetabulum when assessed using obturator oblique radiographic imaging.

### Methods

Computed tomography (CT) scans were performed on the right acetabulum of 50 healthy adults to measure the angle (α) between the posterior wall of the acetabulum and the sagittal plane at the level of the femoral head's maximal diameter. In addition, five cadaveric pelvises were subjected to C-arm fluoroscopic imaging. A 6 cm long, 1.5 mm Kirschner wire was positioned along the posterior surface of the acetabular posterior column, aligned with the greater sciatic notch, and imaged in both the 45° and α-degree obturator oblique views. The radiographic line visualized from the Kirschner wire in the obturator oblique view was defined as the *posterior iliac line*, and its anatomical relationship with the posterior surface of the posterior column was analyzed. Subsequently, a 2.5 mm Kirschner wire was inserted into the posterior column at the standard entry point for screw placement using an electric drill, with the wire tip intentionally positioned between the posterior iliac line and the posterior rim in the 45° obturator oblique view. The trajectory of the wire was assessed under both 45° and α-degree obturator oblique views to determine its relation to the osseous corridor.

### Results

The measured angle between the posterior surface of the acetabular posterior column and the sagittal plane was (60.2±2.5)°. In the 45° obturator oblique view, the

**Data availability statement:** All relevant data are within the manuscript and its Supporting information files.

**Funding:** This work was supported by the National Natural Science Foundation of China (No. 82302031), the Natural Science Foundation of Shandong Province (No. ZR2024QH033), the Shandong Provincial Medical and Health Science and Technology Development Program (202204071124), Weifang City Science and Technology Development Plan (NO.2022YX007 and NO.2024GX064), and Weifang City Health Commission Research Project Plan (WFWSJK-2023-028).

**Competing interests:** The authors have declared that no competing interests exist.

posterior iliac line corresponded with the outer edge of the iliac crest superiorly and the outer edge of the ischium inferiorly, while the posterior wall was projected posterior to the midpoint of the posterior iliac line. In the α° obturator oblique view, the posterior iliac line maintained this alignment but intersected centrally with the posterior acetabular wall. The 2.5 mm Kirschner wire remained within the osseous corridor under the 45° view but potentially extended beyond it under the α° view.

## Conclusion

When the posterior column screw is visualized posterior to the posterior iliac line in the 45° obturator oblique view, further assessment using a α° view is necessary. If the screw appears anterior to the posterior iliac line in the α° view, it indicates that the posterior cortical surface has not been breached.

## Introduction

In clinical practice, acetabular fractures involving both columns account for over 50% of cases, with a considerable proportion amenable to reduction and internal fixation via a single anterior ilioinguinal approach [1–3]. Antegrade posterior column screw fixation is commonly employed for stabilizing posterior column fractures. However, the osseous corridor of the posterior column is narrow, posing a significant risk of screw misplacement beyond the safe trajectory [4,5]. Intraoperatively, fluoroscopic guidance using a C-arm X-ray system facilitates screw placement. The iliac line on the anteroposterior pelvic view is utilized to assess potential penetration of the pelvic inner cortex, while the iliac oblique view helps determine intra-articular screw placement. Nevertheless, evaluating whether the screw breaches the posterior surface of the posterior column remains challenging under the conventional 45° obturator oblique view.

Given the proximity of the sciatic nerve to the posterior column, screw perforation through its posterior cortex carries a risk of iatrogenic nerve injury [6–8]. Therefore, it is critical to investigate the radiographic relationship between posterior column screws and the posterior cortical surface under obturator oblique views.

In this study, computed tomography (CT) scans of the acetabula were performed on 50 healthy adults, and fluoroscopic imaging of five cadaveric pelvises was conducted using a C-arm X-ray system. The objective was to evaluate whether posterior column screws violate the posterior surface of the column when assessed using various obturator oblique imaging angles.

## Materials and methods

This study was approved by the Ethics Committee of Weifang People's Hospital, Shandong Province (Approval No. KYL20240313−1), and all participants provided written informed consent. No personally identifying data were collected. Participation was completely voluntary.

## Study materials

From May 1, 2024, to December 31, 2024, pelvic CT data from 50 healthy adults were collected in the Department of Radiology at Weifang People's Hospital. The right acetabulum was selected for analysis. Cases with fractures, congenital deformities, or other pelvic pathologies were excluded. The participants ranged in age from 25 to 74 years, with a mean age of 38.5 years; there were 26 males and 24 females.

In addition, five dry adult pelvic specimens (three male and two female) provided by the Department of Anatomy of Shandong Second Medical University were used in this study. These specimens were subjected to fluoroscopic imaging in the operating room of Weifang People's Hospital. During fluoroscopy, all pelvic specimens were positioned supine on a radiolucent operating table to simulate the standard positioning for the anterior approach to acetabular surgery. Pelvic alignment was standardized by ensuring that the plane formed by both anterior superior iliac spines and the pubic symphysis was horizontal and parallel to the floor. This positioning is consistent with the typical clinical posture used for anterior approach internal fixation of acetabular fractures.

All fluoroscopic examinations were performed by the same experienced radiologic technologist to ensure consistency in C-arm operation and image acquisition. With accurate alignment of the central X-ray beam, obturator oblique views at 45° and at the predefined α° angle were obtained. For each specimen, images at each angle were acquired at least twice to confirm reproducibility. No apparent positional deviations or significant radiographic variations were observed.

## Imaging techniques and measurements

CT scans were performed using the following parameters: tube voltage 120 kV, tube current 260 mAs, field of view (FOV) 380 mm, slice thickness of 0.6 mm for the bone window and 1.0 mm for the soft tissue window, with a matrix size of 512×512. All subjects were scanned in the supine position. On axial CT images, the posterior surface of the acetabular posterior column appears as a planar structure and is visualized as a straight line. Measurements were taken at the level of the maximal diameter of the femoral head, corresponding to the narrowest portion of the posterior column's safe osseous corridor [9,10]. At this level, the angle (α) between the posterior wall of the acetabulum and the sagittal plane was recorded (Fig 1).

For cadaveric fluoroscopic imaging, a mobile C-arm X-ray system was used. A 1.5 mm diameter, 6 cm long Kirschner wire was positioned along the posterior surface of the posterior column, aligned with the greater sciatic notch (Fig 2). Fluoroscopy was performed at both 45° and α-degree obturator oblique views. The line visualized by the Kirschner wire under these views was defined as the *posterior iliac line* [11,12].

A 2.5-mm Kirschner wire was inserted through the standard entry point for posterior column screw placement. Under the 45° obturator oblique fluoroscopic view, the wire tip was deliberately positioned posterior to the "posterior iliac line" and anterior to the posterior acetabular rim, corresponding to the defined maximal hazardous zone. A stepwise fluoroscopic assessment was then performed. Step 1 (45° view): If the simulated screw (Kirschner wire) was visualized posterior to the posterior iliac line, posterior cortical breach could not be excluded, and further confirmation was required. Step 2 (α-angle view, approximately 60°): The fluoroscope was adjusted to the α-angle obturator oblique view. If the screw image shifted anterior to the posterior iliac line, the screw was considered confined within the safe osseous corridor without posterior cortical penetration. Conversely, if it remained posterior to the line, penetration of the posterior cortex was indicated.

All CT and fluoroscopic image measurements were independently performed by two experienced orthopedic surgeons (Observer A and Observer B) who were blinded to each other's results. Interobserver agreement was assessed using the intraclass correlation coefficient (ICC).

## Results

### Pelvic CT measurement results

The measurements demonstrated excellent interobserver agreement, with an ICC of 0.998(95% confidence interval: 0.997–0.999). The mean value of the angle between the posterior acetabular column/posterior wall surface and the

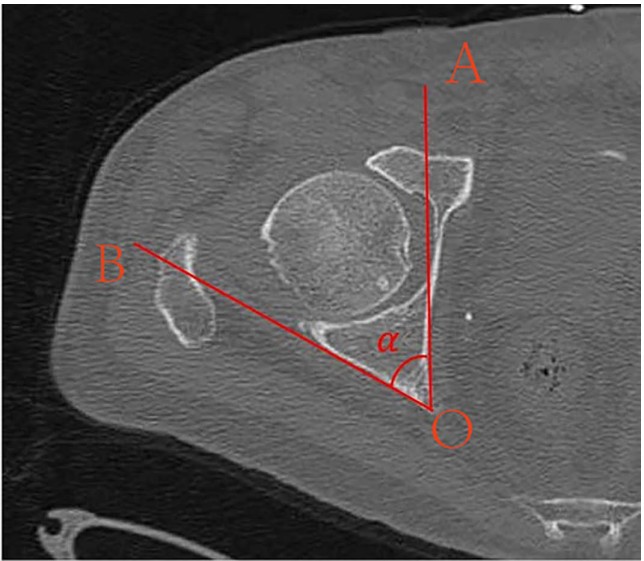

**Fig 1. Axial pelvic CT image at the level of the maximal diameter of the femoral head showing the angle (α) between the posterior wall of the acetabulum and the sagittal plane.** Line OA, parallel to the line connecting the pubic symphysis and the sacral promontory, represents the sagittal reference line. Line OB denotes the tangential line to the posterior acetabular wall.

sagittal plane (α) was 60.2±2.5°, with values ranging from 57.3° to 63.5° (minimum, 57.3°; maximum, 63.5°). Therefore, adding 15° to the standard 45° obturator oblique view—yielding a α° obturator oblique angle—provides a tangential projection of the posterior surface of the posterior column, enhancing the visualization of potential cortical breaches by posterior column screws. At the 45° obturator oblique view, a blind zone (∠AOB) is created on the posterior surface due to overlap from the projected posterior wall of the acetabulum (Fig 3), limiting the accuracy of assessing posterior cortical integrity.

### Fluoroscopic results of cadaveric specimens

In the 45° obturator oblique view, the posterior iliac line aligns superiorly with the outer cortex of the ilium and inferiorly with the lateral margin of the ischium. The projected posterior wall appears posterior to the midpoint of the posterior iliac line. The area posterior to the posterior iliac line was defined as the *maximum danger zone* (D), representing the region at highest risk for screw breach (Fig 4). In the α° obturator oblique view, the posterior iliac line maintains its alignment with the outer iliac and ischial cortices; however, it now becomes tangential to the posterior wall of the acetabulum, aligning centrally with its image (Fig 5).

In all five cadaveric specimens, when the tip of the 2.5-mm Kirschner wire was positioned within the maximal hazardous zone defined on the 45° obturator oblique view—namely, posterior to the posterior iliac line but anterior to the posterior acetabular rim—fluoroscopic images consistently demonstrated that the wire remained within the osseous corridor (Fig 6). However, upon switching to the α° obturator oblique view, all specimens (100%) clearly showed the wire tip located posterior to the posterior iliac line (Fig 7), indicating penetration of the posterior cortical wall of the posterior column (Fig 8). This finding was fully consistent across all specimens.

In the cadaveric experiment, three independent Kirschner wire insertions (simulating posterior column screws) were performed in each of the five specimens, yielding a total of 15 insertions. All insertions followed the standard entry point, and under the 45° obturator oblique view the wire tip was deliberately positioned within the defined "maximal hazardous zone" (between the posterior iliac line and the posterior acetabular rim). In all 15 insertions, fluoroscopic assessment

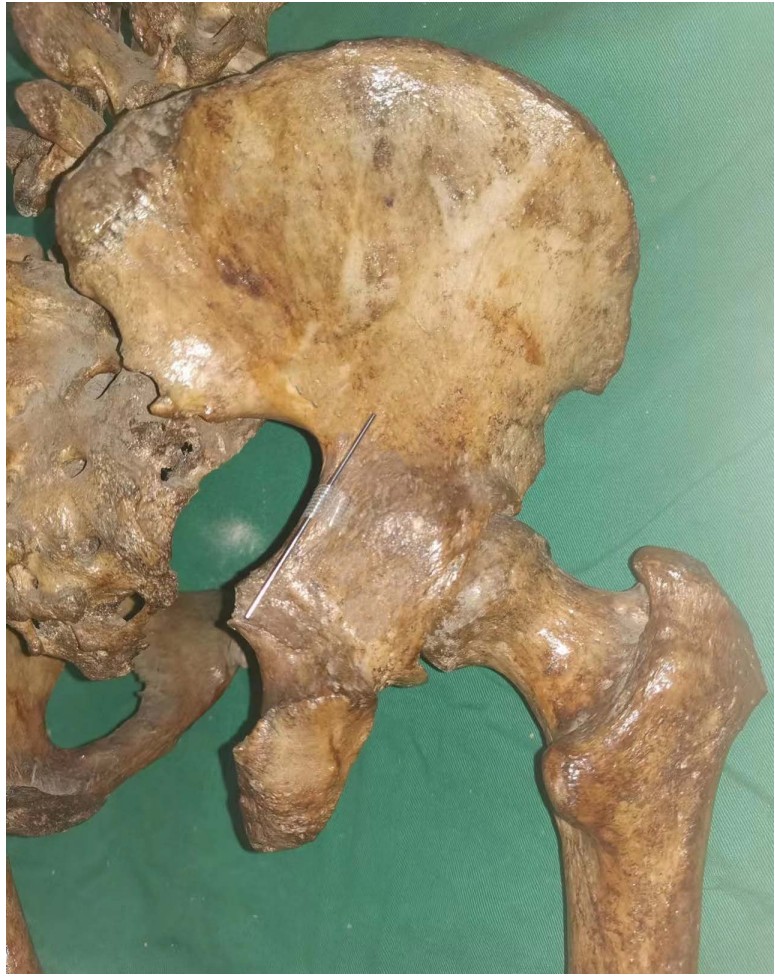

**Fig 2. External view of a right hemipelvis cadaveric specimen showing a 1.5 mm, 6 cm-long Kirschner wire positioned along the posterior surface of the acetabular posterior column at the level of the greater sciatic notch.**

suggested that the Kirschner wire remained within the osseous corridor, with no apparent posterior cortical breach. However, anatomical verification confirmed posterior cortical penetration in all cases. Thus, the 45° view failed to detect posterior breach in 12 of 15 insertions, corresponding to a false-negative rate of 80%, reflecting the presence of a consistent fluoroscopic blind zone. In contrast, all 15 insertions were clearly visualized posterior to the posterior iliac line on the α-angle view, fully consistent with the anatomical findings (100% posterior cortical breach). Accordingly, the diagnostic accuracy of the α-angle view was 100% (15/15), with no false-positive or false-negative results. For all 15 insertions, the assessments based on the 45° and α-angle views were completely discordant: the 45° view suggested a safe position, whereas the α-angle view correctly identified posterior cortical penetration. This 100% discrepancy rate, consistently confirmed by anatomical validation, demonstrates that the α-angle obturator oblique view effectively corrects misinterpretation inherent to the conventional 45° view.

Thus, if the 2.5 mm Kirschner wire (or by inference, a posterior column screw) appears anterior to the posterior iliac line in the α° obturator oblique view, it can be reliably interpreted as being contained within the posterior column without

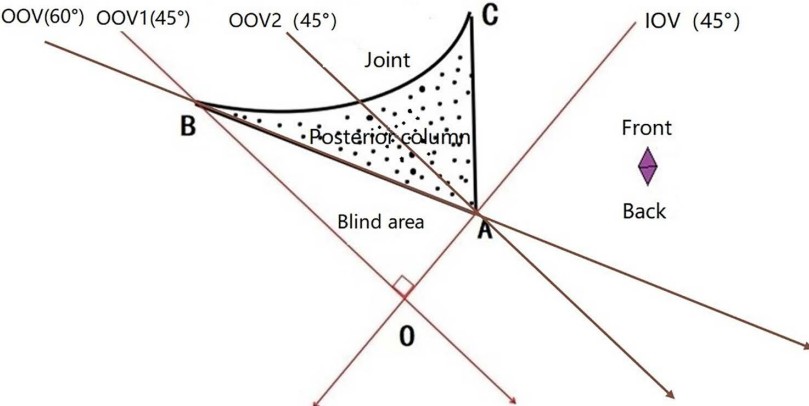

**Fig 3. Schematic diagram of the right acetabular axial CT view at the level of the maximal diameter of the femoral head.** Points A, B, and C define the acetabular posterior wall. Three standard 45° imaging planes are illustrated: obturator oblique view 1 passes through point B, obturator oblique view 2 passes through point A, and the iliac oblique view passes through point A. All three views form a 45° angle with the horizontal plane. Obturator oblique view 1 and the iliac oblique view intersect perpendicularly at point O. A α° obturator oblique view, forming a 30° angle with the horizontal plane, passes through the line connecting points A and B. Points A, O, and B define a radiographic blind zone.

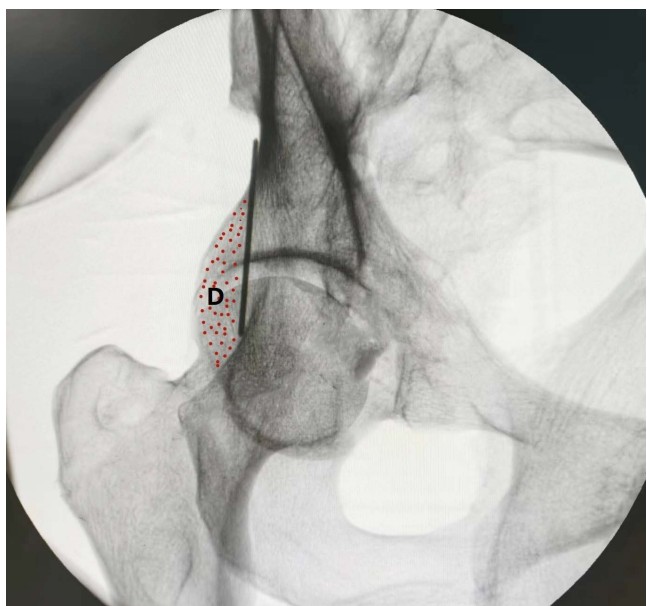

**Fig 4. Under 45° obturator oblique view fluoroscopy, the Kirschner wire projects as the posterior ilioischial line.** This line corresponds superiorly to the outer cortex of the ilium and inferiorly to the outer margin of the ischium. The image of the acetabular posterior wall is located posterior to the midpoint of the posterior ilioischial line. The region posterior to the posterior ilioischial line at this level is defined as the maximum danger zone (D).

breaching its posterior cortex. Conversely, if the wire is visualized posterior to the posterior iliac line in the α° view, it suggests that the screw has penetrated the posterior cortical surface of the posterior column.

## Discussion

Posterior column involvement is frequently observed in acetabular fractures, including both-column, transverse, T-type, and anterior column with posterior hemitransverse fracture patterns [13,14]. In many such cases, a single anterior

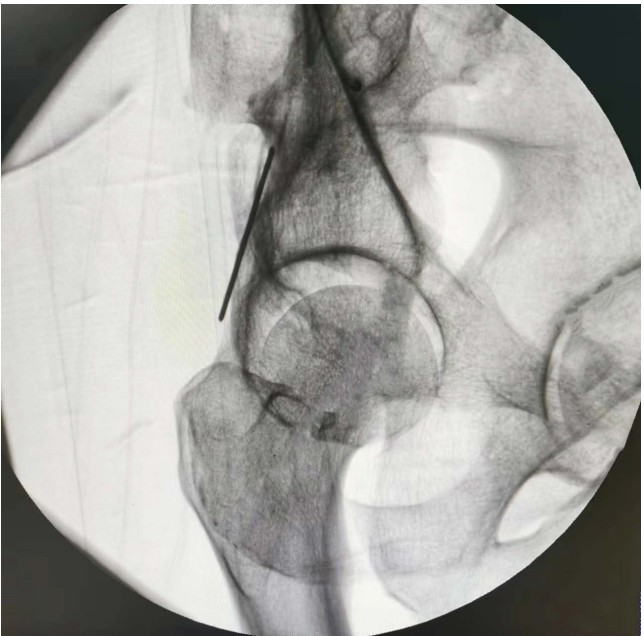

**Fig 5. Under α° obturator oblique view fluoroscopy, the Kirschner wire projects as the posterior ilioischial line.** This line overlaps superiorly with the outer cortex of the ilium, inferiorly with the outer margin of the ischium, and centrally aligns with the acetabular posterior wall. At this angle, the posterior ilioischial line is tangent to the posterior surface of the posterior column.

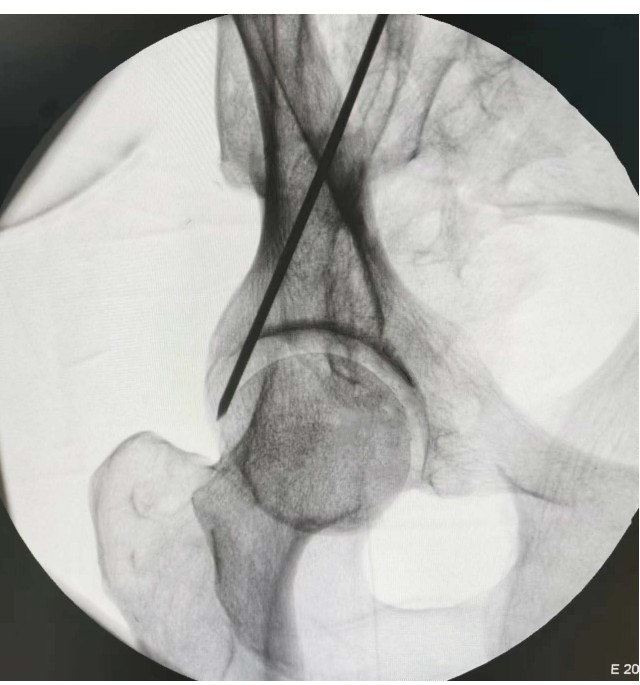

**Fig 6. Under 45° obturator oblique view fluoroscopy, the tip of the 2.5 mm Kirschner wire is located within the maximum danger zone.**

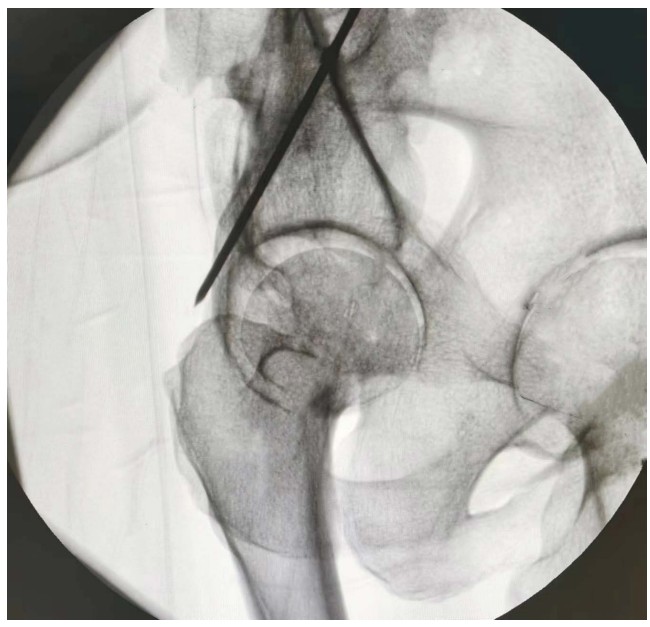

**Fig 7. Under α° obturator oblique view fluoroscopy, the tip of the 2.5 mm Kirschner wire is positioned posterior to the posterior ilioischial line.**

ilioinguinal approach suffices for fracture reduction, during which posterior column screws are commonly employed for internal fixation [15–17]. This technique not only provides reliable mechanical stability, but also offers biomechanical advantages comparable to or exceeding those of posterior plating [18,19]. Furthermore, posterior column screw fixation is less invasive, facilitates central fixation, and eliminates the need for additional posterior incisions—thereby reducing operative trauma, blood loss, and the risk of heterotopic ossification.

The posterior column's osseous corridor is narrow and bordered by critical neurovascular structures. Any cortical breach, particularly posteriorly, carries a high risk of sciatic nerve injury. Despite advancements, intraoperative application of computer navigation and robotic-assisted screw placement remains limited in clinical practice [20–22]. Surgeons still rely heavily on C-arm fluoroscopy, which places a premium on the accuracy of intraoperative imaging. This study aimed to explore the utility of both 45° and α° obturator oblique views in evaluating whether posterior column screws remain within the safe bony channel, using the posterior iliac line as a key radiographic landmark.

The posterior iliac line is defined as the fluoroscopic projection of the posterior surface of the acetabular posterior column as visualized on obturator oblique views at 45° and at the α angle (approximately 60°). On the 45° obturator oblique view, the posterior iliac line is formed by a continuous line connecting the fluoroscopic projection of the outer cortical surface of the ilium superiorly and the outer margin of the ischium inferiorly, representing the posterior boundary of the posterior column corridor. On the α-angle obturator oblique view, the posterior iliac line corresponds to the tangential projection of the posterior acetabular wall. Superiorly, it connects with the projection of the outer iliac cortex, and inferiorly, it merges with the projection of the outer margin of the ischium. When a posterior column screw is visualized posterior to the posterior iliac line on the 45° obturator oblique view, posterior cortical breach cannot be excluded and further assessment is required. If, on the α-angle obturator oblique view, the screw is visualized anterior to the posterior iliac line, it can be confirmed that the screw has not penetrated the posterior surface of the posterior column.

Based on our findings, the 45° and α° (approximately 60°) obturator oblique views play complementary and sequential roles in intraoperative screw assessment. The 45° view can be used as an initial screening tool: if the screw is clearly visualized anterior to the posterior iliac line, it is likely confined within the safe osseous corridor.

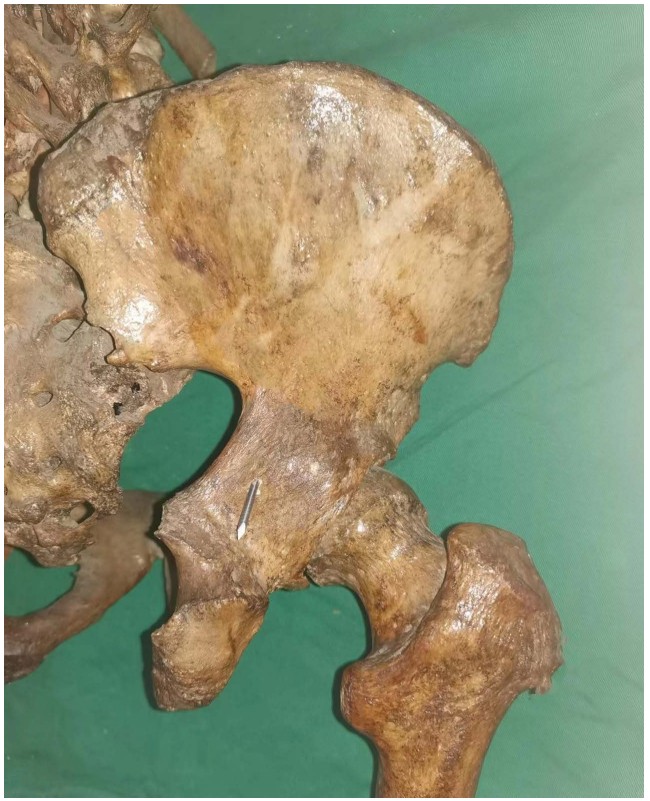

**Fig 8. The tip of the 2.5 mm Kirschner wire is located outside the bony corridor, having penetrated the posterior surface of the posterior column.**

However, when the screw lies within or beyond the defined "maximal hazardous zone" (posterior to the posterior iliac line but anterior to the posterior acetabular wall), the presence of a fluoroscopic blind zone prevents reliable exclusion of posterior cortical breach using the 45° view alone. To illustrate the clinical relevance of this limitation, we reviewed a representative case treated before adoption of the present method. In this patient undergoing anterior fixation for an acetabular fracture, a posterior column screw was judged intraoperatively to be safe based on the standard 45° obturator oblique view, as the screw appeared anterior to the posterior iliac line (Fig 9A). Postoperative CT, however, clearly demonstrated penetration of the posterior cortex of the posterior column, although no sciatic nerve injury occurred (Fig 9B, C). In this context, the α° view serves as a confirmatory assessment. Because this projection is tangential to the posterior cortical surface of the posterior column, the posterior iliac line represents the true outline of the posterior cortex. Accordingly, a screw visualized anterior to this line can be confirmed as intraosseous, whereas a screw located posterior to the line indicates posterior cortical breach.

In both the 45° and α° obturator oblique fluoroscopic views, the posterior iliac line consistently appears, defined by the aggregation of anatomical points corresponding to point A in Fig 3. However, only in the standard 45° obturator oblique view do the *blind zone* and *maximum danger zone* become evident. The blind zone is caused by superimposition of the posterior acetabular wall, creating an occluded area (∠AOB in Fig 3) on fluoroscopy. On CT, the maximum danger zone extends between the entry points of screws visualized under 45° views 1 and 2, encompassing a larger area than the blind zone. On X-ray, the maximum danger zone is demarcated between the posterior iliac line and the posterior acetabular margin (zone D in Fig 4). Therefore, when a posterior column screw is visualized within the maximum danger zone

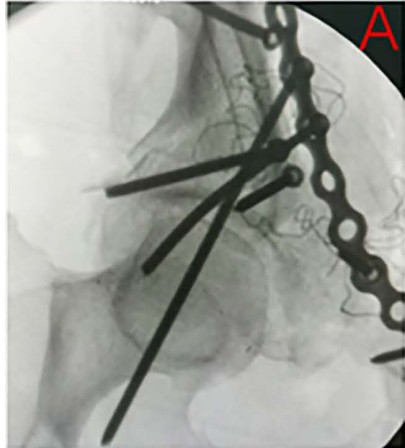 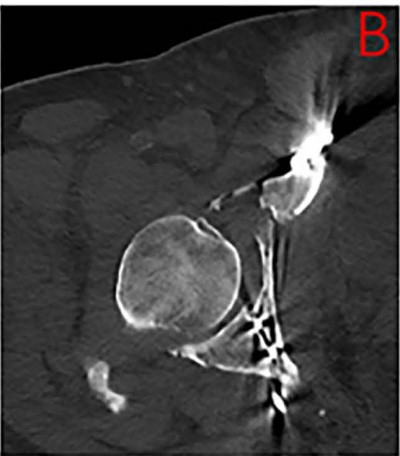 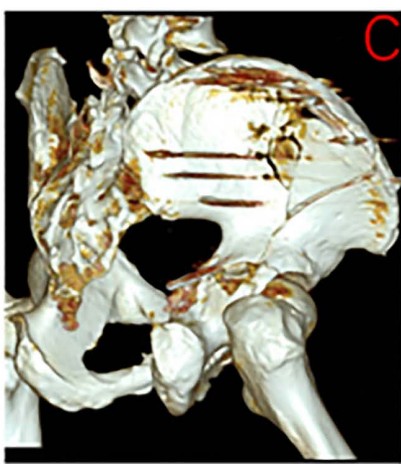

**Fig 9. Screw Penetration of the Posterior Acetabular Wall: A Case of Fluoroscopic Under-Detection. (A)** Intraoperative 45° obturator oblique fluoroscopic view showing no apparent breach of the posterior acetabular wall. **(B, C)** Postoperative CT images demonstrating penetration of the posterior acetabular wall by the screw.

in the 45° view, it cannot be definitively concluded that the screw remains intraosseous—it may have already exited the posterior cortex, necessitating verification via a α° obturator oblique view.

Previous studies by Boni et al.[23] and Yu et al.[18] have emphasized that the α° obturator oblique view provides improved visualization of the posterior surface of the posterior column. However, these studies did not explicitly characterize the radiographic relationship of the posterior iliac line across the two oblique angles, which is a focus of the current investigation. The α° obturator oblique view offers a tangential perspective of the posterior surface of the acetabular posterior column, eliminating the presence of both the blind zone (∠AOB) and the maximum danger zone (D). In this view, if the posterior column screw is located posterior to the posterior iliac line, it can be concluded with high confidence that it has breached the posterior cortical surface. Conversely, if the screw remains anterior to the posterior iliac line at α°, it indicates containment within the osseous channel. Krappinger et al.[24], in their study on posterior wall fixation with screws, reported that a 45° obturator oblique view allows identification of posterior wall penetration when the screw tip is located peripherally. However, if the tip is closer to the inner edge of the wall, a larger viewing angle is required to detect cortical breach—corroborating our findings that a α° view is more appropriate for comprehensive screw trajectory assessment [25,26].

This study has several limitations. First, the cadaveric sample size was small (n = 5), and all specimens were dry pelves lacking soft tissues such as muscles, nerves, and vessels. While this facilitated the identification of radiographic landmarks, it did not replicate intraoperative conditions, including soft-tissue retraction and bleeding, nor did it allow direct assessment of the risk to adjacent structures, such as the sciatic nerve, in cases of screw penetration. Second, the Kirschner wire trajectories used to define the hazardous zone were standardized. Although based on established entry points, screw trajectories in clinical practice may vary according to fracture morphology, quality of reduction, and surgeon preference. Third, both the CT cohort and cadaveric specimens were obtained from a single geographic region. Ethnic and interindividual variations in pelvic morphology may therefore limit the generalizability of the proposed "α° obturator oblique view" and the "posterior iliac line" as universal radiographic landmarks [27–29]. Caution is thus required when applying this method to other populations or complex fracture patterns. Finally, validation was performed under simulated conditions, and prospective intraoperative data are lacking. Future studies should include multicenter anatomical investigations in diverse populations, individualized surgical planning based on preoperative CT, and prospective clinical trials to

confirm the clinical utility of this stepwise fluoroscopic assessment protocol. These efforts will be essential for establishing a standardized workflow and supporting broader clinical application.

The α-angle–based adjustment of the intraoperative obturator oblique fluoroscopic view must account for fracture-related anatomical distortion. In acetabular fractures, displacement, swelling, or hematoma may alter the morphology of the posterior column or posterior wall, limiting accurate α-angle measurement on the injured side. Therefore, when the contralateral acetabulum is intact, its α angle may serve as a reliable reference for intraoperative C-arm angle adjustment. This approach was supported by our data. In 50 healthy adults, the mean α angle did not differ significantly between the left (60.2±2.5.°) and right sides (60.2±2.5°, p>0.05). No significant bilateral asymmetry was observed in the cadaveric specimens. These findings indicate good bilateral consistency of the posterior column anatomy in individuals without structural abnormalities, supporting the use of the contralateral α angle for intraoperative guidance. In cases of bilateral injury or marked anatomical asymmetry, individualized evaluation using preoperative three-dimensional CT reconstruction is recommended, and navigation or robotic assistance may be considered to improve screw placement safety.

It is important to note that the angle between the posterior wall tangent and the sagittal plane is not uniformly α° among individuals. Minor inter-individual variations exist, highlighting the importance of preoperative measurement of this angle using axial CT images. Such preoperative planning allows for personalized adjustment of intraoperative C-arm positioning to ensure accurate radiographic evaluation.

## Supporting information

**S1 File. Original data.**
(XLSX)

## Author contributions

**Conceptualization:** Hongtao Li, Xuecheng Sun, Naibo Feng.

**Data curation:** Hongtao Li, Li Xu, Longxin An, Xiaojing Li.

**Formal analysis:** Hongtao Li, Linjing Zhang.

**Funding acquisition:** Naibo Feng.

**Investigation:** Jun Liu, Naibo Feng.

**Methodology:** Hongtao Li, Kaili Zhai, Xuecheng Sun.

**Project administration:** Jun Liu, Kaili Zhai, Xuecheng Sun, Naibo Feng.

**Software:** Kaili Zhai.

**Supervision:** Hongtao Li.

**Writing – original draft:** Hongtao Li.

**Writing – review & editing:** Naibo Feng.

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
