## [Decision Letter · Decision Letter 0]

20 Dec 2025

Thank you for submitting your manuscript to PLOS ONE. After careful consideration, we feel that it has merit but does not fully meet PLOS ONE’s publication criteria as it currently stands. Therefore, we invite you to submit a revised version of the manuscript that addresses the points raised during the review process.

This is an interesting and clinically relevant study in which the authors have evaluated the utility of obturator oblique radiographic views in assessing the relationship between screw trajectories and the posterior cortical surface of the acetabulum. After thorough consideration of the reviewers’ comments and an overall assessment of the manuscript’s quality, the editorial decision is: **MAJOR REVISION**

We look forward to receiving your revised manuscript.

Kind regards,

Richa Gupta

Academic Editor

PLOS One

Journal Requirements:

1. Please ensure that your manuscript meets PLOS ONE’s style requirements, including those for file naming. The PLOS ONE style templates can be found at

4. We note that your Data Availability Statement is currently as follows: All relevant data are within the manuscript and its Supporting Information files

Additional Editor Comments:

EDITORIAL DECISION

This is an interesting and clinically relevant study in which the authors have evaluated the utility of obturator oblique radiographic views in assessing the relationship between screw trajectories and the posterior cortical surface of the acetabulum. After thorough consideration of the reviewers’ comments and an overall assessment of the manuscript’s quality, the editorial decision is: MAJOR REVISION

Please find attached reviewer’s comments:

REVIEWER 1 – MAJOR REVISION

The manuscript investigates whether posterior column screws penetrate the posterior cortical surface of the acetabulum when assessed using obturator oblique radiographs. The topic is clinically significant, as it directly relates to the intraoperative safety of acetabular fracture fixation.

1.The method used to measure the α angle should be described in more detail and clearly illustrated in a figure. It is not specified which anatomical reference points were used to define the sagittal plane (e.g., was it based on the line connecting the symphysis pubis and the sacral promontory?).

2. In the cadaveric study, the pelvic positioning and reproducibility of fluoroscopic imaging should be clarified. Cadaveric specimens may differ from living patients in terms of anatomical posture, which could affect measurement accuracy.

3. If multiple observers performed the measurements, interobserver reliability should be reported.

4. Demographic data of the CT cohort appear incomplete; the sex distribution (number of male and female participants) of the 50 healthy subjects should be specified.

5. The study appears to have been conducted in a single ethnic population, which may limit the generalizability of the α angle. This limitation should be acknowledged and discussed.

6. In the cadaveric analysis, the authors report that the Kirschner wire appeared intraosseous at 45° but breached the posterior cortex at 60°. It should be clarified whether this finding was consistent across all five specimens or observed only in selected cases, as this affects the generalizability of the results.

7. Beyond the mean and standard deviation, no additional statistical data are provided. It is recommended to include the minimum and maximum values of the measured α angles to better reflect the variability of the dataset.

8. The authors suggest that preoperative measurement of the α angle may guide intraoperative imaging. However, in fracture cases, the posterior column or wall anatomy may be distorted, making calculation on the injured side unreliable.Do the authors recommend measuring the α angle from the contralateral (uninjured) side?

If so, was any right–left difference assessed (e.g., bilateral comparison within the same individuals or supported by literature data)?

REVIEWER 2 – MINOR REVISION

Clarify whether the study included both male and female specimens, as pelvic morphology varies.

Important methodological details are included, but the narrative becomes overly technical. Consider simplifying the description of the imaging setup and experimental steps.

The rationale for measuring CT angles (α) and how this measurement informs the cadaveric imaging is not clearly explained.

It is unclear whether α refers to an individualized or an average angle - cadaveric work uses 60°, but the sample average is 60.2°, suggesting generalization. Clarify whether each pelvis was imaged at its own α or at a standardized angle.

The term posterior iliac line is newly defined here; however, its anatomical and clinical significance needs clearer justification.

The results describe radiographic relationships but do not clearly quantify the method’s accuracy or reliability. Author/s may add the number of cases in which screw placement assessment differed between 45° and α/60° views and whether observer agreement was assessed.

Author/s may add how the 45° and 60° views complement each other and what the clinical implication is (i.e., improved intraoperative safety).

Author/s may add limitations such as small cadaver sample sizes, the absence of soft tissue, standardized wire trajectories, and variability in pelvic morphology, which may affect generalizability to clinical practice.

Author/s may add future studies with larger samples, bilateral evaluation, and clinical validation to confirm the reliability and generalizability of this radiographic assessment technique.

REVIEWER 3 – ACCEPT

Interesting study of integrating x rays in oblique views for visualizing posterior column screws. Hope this becomes useful in clinical practice. The analysis is well done and the article is well written.

Reviewers' comments:

Reviewer’s Responses to Questions

**Comments to the Author**

1. Is the manuscript technically sound, and do the data support the conclusions?

Reviewer #1: Yes

Reviewer #2: Yes

Reviewer #3: Yes

2. Has the statistical analysis been performed appropriately and rigorously?

Reviewer #1: Yes

Reviewer #2: N/A

Reviewer #3: Yes

3. Have the authors made all data underlying the findings in their manuscript fully available?

Reviewer #1: Yes

Reviewer #2: Yes

Reviewer #3: Yes

4. Is the manuscript presented in an intelligible fashion and written in standard English?

Reviewer #1: Yes

Reviewer #2: Yes

Reviewer #3: Yes

Reviewer #1: The manuscript investigates whether posterior column screws penetrate the posterior cortical surface of the acetabulum when assessed using obturator oblique radiographs. The topic is clinically significant, as it directly relates to the intraoperative safety of acetabular fracture fixation.

1.The method used to measure the α angle should be described in more detail and clearly illustrated in a figure. It is not specified which anatomical reference points were used to define the sagittal plane (e.g., was it based on the line connecting the symphysis pubis and the sacral promontory?).

2. In the cadaveric study, the pelvic positioning and reproducibility of fluoroscopic imaging should be clarified. Cadaveric specimens may differ from living patients in terms of anatomical posture, which could affect measurement accuracy.

3. If multiple observers performed the measurements, interobserver reliability should be reported.

4. Demographic data of the CT cohort appear incomplete; the sex distribution (number of male and female participants) of the 50 healthy subjects should be specified.

5. The study appears to have been conducted in a single ethnic population, which may limit the generalizability of the α angle. This limitation should be acknowledged and discussed.

6. In the cadaveric analysis, the authors report that the Kirschner wire appeared intraosseous at 45° but breached the posterior cortex at 60°. It should be clarified whether this finding was consistent across all five specimens or observed only in selected cases, as this affects the generalizability of the results.

7. Beyond the mean and standard deviation, no additional statistical data are provided. It is recommended to include the minimum and maximum values of the measured α angles to better reflect the variability of the dataset.

8. The authors suggest that preoperative measurement of the α angle may guide intraoperative imaging. However, in fracture cases, the posterior column or wall anatomy may be distorted, making calculation on the injured side unreliable.Do the authors recommend measuring the α angle from the contralateral (uninjured) side?

If so, was any right–left difference assessed (e.g., bilateral comparison within the same individuals or supported by literature data)?

Reviewer #2: Clarify whether the study included both male and female specimens, as pelvic morphology varies.

Important methodological details are included, but the narrative becomes overly technical. Consider simplifying the description of the imaging setup and experimental steps.

The rationale for measuring CT angles (α) and how this measurement informs the cadaveric imaging is not clearly explained.

It is unclear whether α refers to an individualized or an average angle - cadaveric work uses 60°, but the sample average is 60.2°, suggesting generalization. Clarify whether each pelvis was imaged at its own α or at a standardized angle.

The term posterior iliac line is newly defined here; however, its anatomical and clinical significance needs clearer justification.

The results describe radiographic relationships but do not clearly quantify the method’s accuracy or reliability. Author/s may add the number of cases in which screw placement assessment differed between 45° and α/60° views and whether observer agreement was assessed.

Author/s may add how the 45° and 60° views complement each other and what the clinical implication is (i.e., improved intraoperative safety).

Author/s may add limitations such as small cadaver sample sizes, the absence of soft tissue, standardized wire trajectories, and variability in pelvic morphology, which may affect generalizability to clinical practice.

Author/s may add future studies with larger samples, bilateral evaluation, and clinical validation to confirm the reliability and generalizability of this radiographic assessment technique.

Reviewer #3: Interesting study of integrating x rays in oblique views for visualizing posterior column screws. Hope this becomes useful in clinical practice. The analysis is well done and the article is well written.

.

Reviewer #1: **Yes:** erdem ateşerdem ateşerdem ateşerdem ateş

Reviewer #2: **Yes:** Dr Swati GoyalDr Swati GoyalDr Swati GoyalDr Swati Goyal

Reviewer #3: No

---

## [Author Response · Author response to Decision Letter 1]

21 Jan 2026

1. Please ensure that your manuscript meets PLOS ONE’s style requirements, including those for file naming. The PLOS ONE style templates can be found at

Response: The modifications have been completed as required.

Response: The modifications have been completed as required.

Response: We apologize for the inconsistency between the previous version of the manuscript and the information provided in the submission system regarding the financial disclosure. This issue has been corrected in the revised version.

This work was supported by the National Natural Science Foundation of China (No. 82302031), the Natural Science Foundation of Shandong Province (No. ZR2024QH033), the Shandong Provincial Medical and Health Science and Technology Development Program (202204071124), Weifang City Science and Technology Development Plan (NO.2022YX007 and NO.2024GX064), and Weifang City Health Commission Research Project Plan (WFWSJK-2023-028).

4. We note that your Data Availability Statement is currently as follows: All relevant data are within the manuscript and its Supporting Information files

Response: The modifications have been completed as required.

Response: Not applicable.

Reviewer #1: The manuscript investigates whether posterior column screws penetrate the posterior cortical surface of the acetabulum when assessed using obturator oblique radiographs. The topic is clinically significant, as it directly relates to the intraoperative safety of acetabular fracture fixation.

Response to Reviewer:

We sincerely thank the reviewer for highlighting the clinical significance of this study. We appreciate the recognition that accurate assessment of posterior column screw placement using obturator oblique radiographs is directly relevant to intraoperative safety in acetabular fracture fixation.

1.The method used to measure the α angle should be described in more detail and clearly illustrated in a figure. It is not specified which anatomical reference points were used to define the sagittal plane (e.g., was it based on the line connecting the symphysis pubis and the sacral promontory?).

Response to Reviewer:

Thank you for this valuable and constructive comment. We fully agree that clearly defining the anatomical reference points for the sagittal plane and providing a detailed description of the α angle measurement are essential to ensure the reproducibility and methodological rigor of the study.

Accordingly, we have revised Figure 1 and expanded the description of the measurement procedure as follows. The α angle is defined as the angle between the tangent to the posterior wall of the acetabulum and the sagittal plane at the axial slice corresponding to the maximal diameter of the femoral head (as shown in the revised Figure 1).

In the revised Figure 1, the following key elements are explicitly illustrated and labeled:

(1) The anatomical locations of the symphysis pubis and the sacral promontory, as well as the line connecting these two landmarks, which serves as the reference for defining the sagittal plane.

(2) Line OA, drawn parallel to the symphysis pubis–sacral promontory line and explicitly labeled as the “sagittal plane reference line.”

(3) Line OB, representing the tangent to the posterior acetabular wall, explicitly labeled as the “posterior acetabular wall tangent.”

(4) The angle formed between lines OA and OB, labeled as the “α angle.”

(5) Anatomical landmarks at the level of the maximal femoral head diameter, including the contour of the femoral head and the boundary of the posterior acetabular column.

Through these revisions, we have clearly specified the anatomical reference standard for the sagittal plane and provided a step-by-step clarification of the α angle measurement method, thereby improving its clarity, operability, and reproducibility. The revised Figure 1 has been incorporated into the manuscript accordingly.

Fig. 1. Axial pelvic CT image at the level of the maximal diameter of the femoral head showing the angle (α) between the posterior wall of the acetabulum and the sagittal plane. Line OA, parallel to the line connecting the pubic symphysis and the sacral promontory, represents the sagittal reference line. Line OB denotes the tangential line to the posterior acetabular wall.

2. In the cadaveric study, the pelvic positioning and reproducibility of fluoroscopic imaging should be clarified. Cadaveric specimens may differ from living patients in terms of anatomical posture, which could affect measurement accuracy.

Response to Reviewer:

We sincerely thank the reviewer for this insightful comment regarding pelvic positioning and the reproducibility of fluoroscopic imaging in the cadaveric study. We fully acknowledge that differences between cadaveric specimens and living patients, particularly in anatomical posture and dynamic conditions, may potentially influence measurement accuracy. Therefore, special attention was paid to standardizing pelvic positioning and ensuring imaging reproducibility in our experimental design.

In this cadaveric study, all pelvic specimens were positioned in the supine position on a radiolucent operating table to simulate the standard intraoperative posture used in anterior acetabular surgical approaches. During positioning, the pelvis was carefully adjusted so that the plane defined by the bilateral anterior superior iliac spines (ASISs) and the symphysis pubis was horizontal and parallel to the ground. This three-point alignment method is consistent with the conventional supine pelvic positioning commonly adopted during anterior acetabular fracture surgery in clinical practice, thereby enhancing the clinical relevance of our fluoroscopic measurements.

All fluoroscopic imaging procedures were performed by the same experienced radiologic technologist to ensure consistency in C-arm manipulation and image acquisition. The C-arm was precisely adjusted to obtain obturator oblique views at angles of 45° and the predefined α° relative to the horizontal plane, while maintaining consistent central beam alignment across all specimens. For each specimen, fluoroscopic images at each angle were acquired at least twice to verify reproducibility. No evident positional displacement or significant image variability was observed between repeated acquisitions.

We acknowledge that cadaveric studies cannot fully replicate the dynamic intraoperative conditions encountered in living patients, such as soft tissue interference, bleeding, or respiratory motion. Nevertheless, through strict control of pelvic positioning and a standardized fluoroscopic imaging protocol, we ensured that the radiographic measurements and the definition of the posterior acetabular line demonstrated good reproducibility and meaningful clinical reference value.

These methodological details have been added to the revised Study Materials section of the manuscript to improve clarity and transparency. (P5L89---P6L104)

3. If multiple observers performed the measurements, interobserver reliability should be reported.

Response to Reviewer:

We sincerely thank the reviewer for this important and constructive suggestion. Reporting interobserver reliability is indeed essential to demonstrate the robustness and scientific rigor of the measurement methodology.

In the present study, measurements of the angle α (defined as the angle between the posterior acetabular column/posterior wall surface and the sagittal plane on pelvic CT images) were independently performed by two orthopedic surgeons who had received standardized training in musculoskeletal imaging assessment. In accordance with the reviewer’s recommendation, we have now evaluated interobserver reliability using the intraclass correlation coefficient (ICC).

The corresponding methodological description and results have been added to the revised manuscript as follows:

1. Materials and Methods – Imaging Techniques and Measurements

The following statement has been added at the end of this subsection:

“All CT and fluoroscopic image measurements were independently performed by two experienced orthopedic surgeons (Observer A and Observer B) who were blinded to each other’s results. Interobserver agreement was assessed using the intraclass correlation coefficient (ICC).” (P7L132---P7L135)

2. Results – Pelvic CT Measurement Results

The following sentence has been added at the beginning of this subsection:

“The measurements demonstrated excellent interobserver agreement, with an intraclass correlation coefficient (ICC) of 0.998 (95% confidence interval: 0.997–0.999). The mean value of the angle between the posterior acetabular column/posterior wall surface and the sagittal plane (α) was 60.2 ± 2.5°.” (P7L138---P7L141)

These additions further strengthen the methodological reliability of our study and address the reviewer’s concern regarding measurement consistency.

4. Demographic data of the CT cohort appear incomplete; the sex distribution (number of male and female participants) of the 50 healthy subjects should be specified.

Response to Reviewer:

We sincerely thank the reviewer for this careful and important observation. We fully agree that providing complete demographic information, including sex distribution, is essential for clearly characterizing the study population, assessing potential sources of bias, and improving the transparency and reproducibility of the research.

In response to this comment, we have supplemented the demographic data of the CT cohort by specifying the sex distribution of the 50 healthy subjects and revised the Materials and Methods section accordingly. The following information has been added to the subsection describing the study materials:

“From May 1, 2024, to December 31, 2024, pelvic CT data from 50 healthy adults were collected in the Department of Radiology at Weifang People’s Hospital. The right acetabulum was selected for analysis. Cases with fractures, congenital deformities, or other pelvic pathologies were excluded. The participants ranged in age from 25 to 74 years, with a mean age of 38.5 years; there were 26 males and 24 females.” (P5L84---P5L88)

These revisions address the reviewer’s concern and provide a more complete description of the study population.

5. The study appears to have been conducted in a single ethnic population, which may limit the generalizability of the α angle. This limitation should be acknowledged and discussed.

Response to Reviewer:

We sincerely thank the reviewer for this insightful and important comment. We fully agree that the fact that this study was conducted within a single ethnic population represents an inherent limitation, which may restrict the generalizability of the measured α angle across different populations. Consideration of population-specific anatomical variation is particularly important when interpreting and applying quantitative morphometric parameters in clinical practice.

In response to this comment, we have added a dedicated paragraph to the Discussion section to explicitly acknowledge and discuss this limitation and its potential impact on the interpretation of our findings. The added text reads as follows,(P13L273---P14L292):

This study has several limitations. First, the cadaveric sample size was small (n = 5), and all specimens were dry pelves lacking soft tissues such as muscles, nerves, and vessels. While this facilitated the identification of radiographic landmarks, it did not replicate intraoperative conditions, including soft-tissue retraction and bleeding, nor did it allow direct assessment of the risk to adjacent structures, such as the sciatic nerve, in cases of screw penetration. Second, the Kirschner wire trajectories used to define the hazardous zone were standardized. Although based on established entry points, screw trajectories in clinical practice may vary according to fracture morphology, quality of reduction, and surgeon preference. Third, both the CT cohort and cadaveric specimens were obtained from a single geographic region. Ethnic and interindividual variations in pelvic morphology may therefore limit the generalizability of the proposed “α° obturator oblique view” and the “posterior iliac line” as universal radiographic landmarks [27-29]. Caution is thus required when applying this method to other populations or complex fracture patterns. Finally, validation was performed under simulated conditions, and prospective intraoperative data are lacking. Future studies should include multicenter anatomical investigations in diverse populations, individualized surgical planning based on preoperative CT, and prospective clinical trials to confirm the clinical utility of this stepwise fluoroscopic assessment protocol. These efforts will be essential for establishing a standardized workflow and supporting broader clinical application.

6. In the cadaveric analysis, the authors report that the Kirschner wire appeared intraosseous at 45° but breached the posterior cortex at 60°. It should be clarified whether this finding was consistent across all five specimens or obs

---

## [Decision Letter · Decision Letter 1]

5 Apr 2026

Utilization of the posterior iliac line for visualizing posterior column screws in obturator oblique view

PONE-D-25-26318R1

Dear Dr. Feng,

We’re pleased to inform you that your manuscript has been judged scientifically suitable for publication and will be formally accepted for publication once it meets all outstanding technical requirements.

Kind regards,

Mohmed Isaqali Karobari, BDS, MScD.Endo, Ph.D. Endo, FDS, FPFA, FICD, MFDS

Academic Editor

PLOS One

Additional Editor Comments (optional):

Dear Authors,

The authors have addressed all the reviewers' comments and suggestions, and the manuscript has undergone significant improvement. The manuscript can be accepted for publication in its current form. I would like to congratulate the authors and wish them all the very best in their future endeavours.

Best regards and keep well

Reviewers' comments:

Reviewer’s Responses to Questions

**Comments to the Author**

Reviewer #1: All comments have been addressed

Reviewer #3: All comments have been addressed

2. Is the manuscript technically sound, and do the data support the conclusions?

Reviewer #1: Yes

Reviewer #3: Yes

3. Has the statistical analysis been performed appropriately and rigorously?

Reviewer #1: Yes

Reviewer #3: Yes

4. Have the authors made all data underlying the findings in their manuscript fully available?

Reviewer #1: Yes

Reviewer #3: Yes

5. Is the manuscript presented in an intelligible fashion and written in standard English?

Reviewer #1: Yes

Reviewer #3: Yes

Reviewer #1: Dear Editor,

I have carefully reviewed the revised version of the manuscript. The authors have adequately addressed the previously raised comments and made the necessary improvements.

In its current form, the manuscript is suitable for publication.

Kind regards,

Reviewer #3: Thank you for the scientific article, The Authors have addressed the clarifications from the reviewers adequately.

.

Reviewer #1: **Yes:** Erdem AteşErdem AteşErdem AteşErdem Ateş

Reviewer #3: **Yes:** Santosh PV RaiSantosh PV RaiSantosh PV RaiSantosh PV Rai

---

## [Editor Report · Acceptance letter]

PONE-D-25-26318R1

PLOS One

Dear Dr. Feng,

I'm pleased to inform you that your manuscript has been deemed suitable for publication in PLOS One. Congratulations! Your manuscript is now being handed over to our production team.

Kind regards,

on behalf of

Prof Dr. Mohmed Isaqali Karobari

Academic Editor

PLOS One